# Limitations of Phage Therapy and Corresponding Optimization Strategies: A Review

**DOI:** 10.3390/molecules27061857

**Published:** 2022-03-13

**Authors:** Jiaxi Lin, Fangyuan Du, Miao Long, Peng Li

**Affiliations:** 1College of Animal Science & Veterinary Medicine, Shenyang Agricultural University, Shenyang 110866, China; 2019240393@stu.syau.edu.cn (J.L.); fangyuandu@126.com (F.D.); longmiao@syau.edu.cn (M.L.); 2Key Laboratory of Ruminant Infectious Disease Prevention and Control (East), Ministry of Agriculture and Rural Affairs, Shenyang 110866, China

**Keywords:** bacteriophage, phage therapy, limitations, applications, existing solutions

## Abstract

Bacterial infectious diseases cause serious harm to human health. At present, antibiotics are the main drugs used in the treatment of bacterial infectious diseases, but the abuse of antibiotics has led to the rapid increase in drug-resistant bacteria and to the inability to effectively control infections. Bacteriophages are a kind of virus that infects bacteria and archaea, adopting bacteria as their hosts. The use of bacteriophages as antimicrobial agents in the treatment of bacterial diseases is an alternative to antibiotics. At present, phage therapy (PT) has been used in various fields and has provided a new technology for addressing diseases caused by bacterial infections in humans, animals, and plants. PT uses bacteriophages to infect pathogenic bacteria so to stop bacterial infections and treat and prevent related diseases. However, PT has several limitations, due to a narrow host range, the lysogenic phenomenon, the lack of relevant policies, and the lack of pharmacokinetic data. The development of reasonable strategies to overcome these limitations is essential for the further development of this technology. This review article described the current applications and limitations of PT and summarizes the existing solutions for these limitations. This information will be useful for clinicians, people working in agriculture and industry, and basic researchers.

## 1. Introduction

### 1.1. Status of Bacterial Resistance

Bacterial infectious diseases seriously endanger human health. According to the World Health Organization, more than 25% of deaths worldwide are caused by infectious diseases, and bacteria account for 38% of human pathogens [1,2]. By the beginning of this century, the number of deaths from bacterial infections in the world had risen to 20 million. At present, antibiotics are mainly used to treat bacterial infectious diseases, but the abuse of antibiotics has led to the rapid increase of drug-resistant bacteria, especially multi-drug-resistant bacteria (bacteria that are simultaneously resistant to three or more kinds of antibiotics used in the clinic). Due to the inability to effectively control infections by multi-drug-resistant bacteria, this has become a serious problem in clinical treatment. Currently, drug-resistant diseases kill about 700,000 people worldwide each year, and if no action is taken, this figure could increase to 10 million a year by 2050 [3]. There are more and more common diseases that cannot be cured, including respiratory tract infections, sexually transmitted infections, and urinary tract infections [4,5]. The resistance of bacteria to antibiotics exerts a heavy economic burden and causes huge health problems in the society; therefore, it is imperative to explore new ways to target multi-drug-resistant bacteria [6]. The emergence of a variety of drug-resistant bacteria has raised interest in alternatives to conventional antimicrobials. One of the possible alternatives to antibiotics are bacteriophages that can be used as antimicrobial agents.

### 1.2. Bacteriophages

Bacteriophage is a specific intracellular virus that infects and engulfs bacteria [7]. Bacteriophages were discovered almost simultaneously by Frederick William Twort in England, and Félix d’Herelle in France [8]. Bacteriophages are the most abundant viral entities on Earth, ubiquitous in all ecosystems. among which, seawater is their preferred natural environment [9]. The genetic material of bacteriophages consists of double-stranded or single-stranded DNA or RNA. Morphologically, bacteriophages can be caudate, polyhedral, filamentous, or pleomorphic. A typical bacteriophage usually contains an icosahedral head, a hollow needle-like structure, a tail consisting of an outer sheath, and a base composed of tail filaments and tail needles [10]. Bacteriophages can have cleavage or lysogen life cycles (Figure 1), the latter of which can be lytic or temperate. Lytic bacteriophages can replicate during the lytic cycle, a process that involves producing new viral offspring and releasing them from infected cells. Temperate phages are those whose DNA is integrated only into the nuclear genome of the host after adsorption and invasion and can be replicated synchronously with the host DNA for a long period. Temperate phages generally do not proliferate and cause host cell lysis in general. Under pressure conditions, temperate phages can exit the lysogenic state and produce more virions that are released from bacteria.

### 1.3. Phage Therapy (PT)

PT uses bacteriophages to infect pathogenic bacteria so to stop bacterial infections and treat and prevent related diseases. The therapeutic potential of bacteriophages against bacterial infections has been long studied, and one of its proponents, Félix d’Herelle, described his experiment with rabbits against *Shigella* in his first paper. Most of the early phage studies that were conducted between the 1920s and the 1930s focused on the development of PT against bacterial infections. Numerous companies then began to sell phage preparations. In 1920, bacteriophages were used to treat human infections in Eastern Europe and the former Soviet Union and achieved good results. In 1921, Bruynoghe and Maisin used bacteriophages to treat skin infections caused by *Staphylococci*. In the late 1930s, however, the Pharmaceutical and Chemical Committee of the American Medical Association concluded that the efficacy of PT was unclear [11]. They acknowledge that there are both positive and negative results in the literature, but they expressed the concern that little was known about the biological properties of bacteriophages and that bacteriophages lacked standardization and standards of purity and potency, which made it impossible to compare most published studies. Further research was clearly needed. These factors and the success of emerging antibiotics in the 1940s resulted in a decline in the interest in PT [11,12]. However, in recent years, the relentless rise in antibiotic resistance has ignited a renewed interest in PT and provided a new impetus for research on bacteriophage-based bacterial infection solutions. The Southeast Asia region has embarked on a phage revolution, paving the way for phage therapy and related policies and addressing the threat posed by multidrug-resistant pathogens in the region and around the world [13].

### 1.4. PT Application

The results of PT research are currently used in various fields (Table 1) to control the growth and proliferation of pathogenic bacteria [14]. PT is used to treat not only human diseases caused by bacteria [15,16,17,18] but also animal and plant diseases of bacterial origin [19,20,21], as well as to ensure food safety [22,23,24]. The applications in human, animal, and plant bacterial diseases and to ensure food safety and quality have significant effects. PT has a lysis effect on a variety of bacteria, including drug-resistant *Staphylococcus aureus* and *Pseudomonas aeruginosa* and pathogenic *Escherichia coli* and *Salmonella*. The table lists the therapeutic effects of different phages or phage mixtures on different bacteria in various fields.

Although bacteriophages have been studied for almost a century, data on the treatment of bacterial infections are incomplete. This was one of the reasons why PT was abandoned for some time. With the in-depth study of PT in recent years, the application of related technologies, and the abundance of collected data, the large-scale application of PT is today feasible [59]. Although PT has broad application prospects, it also has inevitable limitations such as a narrow host spectrum, immune clearance by the body, and emergence of anti-phage bacterial strains. Adopting reasonable strategies to overcome these limitations through an in-depth understanding of the properties of phages and of their impact on the host is a priority for the further development of this burgeoning industry. This paper starts describing the problems existing in PT, introduces in detail the limitations in the application of PT, and summarizes the current strategies to overcome these limitations. It will provide a reference for clinicians, people working in agriculture and industry, and basic researchers and will lay a foundation for the development of PT.

## 2. Limitations of PT

### 2.1. Disadvantageous Characteristics of Bacteriophages

The cleavage spectrum of bacteriophages is too narrow because of its high specificity. Bacteriophages usually act only on certain genera of bacteria, some on a limited number of species, and thus, cannot target all pathogenic strains of a single bacterial species [60]. Bacteriophages are useful in managing diseases caused by a single bacterium, but the actual clinical cases are often infections that are caused by a variety of pathogenic bacteria. Hence, it is often difficult for specific bacteriophages to have a desired therapeutic effect [21]. The lysogenic phenomenon consist in the fact that some lysogenic phages cannot lyse the host bacteria and inhibit the lytic effect of other phages on their host bacteria after integration with the host bacteria. In lysogenicity, the viral genome replicates with the host DNA, either in a free plasmid-like state or after integration into the bacterial chromosome [13]. In addition, a more important problem is that bacteriophages in the lysogenic state can also transmit toxins and antibiotic resistance genes to bacteria.

In contrast to protein drugs whose activity and purity can be assessed based on specific antibodies titers, the composition of PT preparations is more complex and includes both proteins and nucleic acids. Thus, it is difficult to evaluate its quality and curative effects [61].

### 2.2. Lack of Relevant Policies

Policies and regulations on the clinical application of PT are lacking [62]. Appropriate regulatory standards can create opportunities to raise awareness of this promising treatment. Verbeken et al. provided a detailed analysis of the opinion of European stakeholders and discussed the need to adjust the regulatory framework to accommodate PT [63]. One important consideration is whether PT development occurs on an industrial scale or on a hospital-based, patient-specific scale. They argued for a new, dedicated European regulatory framework for PT. In addition, there is no clear standard for phage isolation and purification, which makes the efficacy of isolated phage preparations variable. There is no standardized procedure in clinical treatments with bacteriophages.

### 2.3. Resistance of Bacteria to Bacteriophages

With the emergence of bacteriophage-resistant strains, several studies found that if a single phage is used repeatedly for a long time, bacteria also evolve phage-resistant strains in the process of natural selection [64,65,66,67]. This is part of a series of anti-bacteriophage strategies evolved in the long-term in bacteria, which include adsorption inhibition, restriction modification systems, injection blocking, abortion infection, superinfection immunity, and the Clustered Regularly Interspaced Short Palindromic Repeats-CRISPR associated (CRISPR–Cas) system [60]. Adsorption resistance results in reduced interactions between bacteriophages and bacteria. In abortion infection, both bacteriophages and bacteria die. CRISPR–Cas is part of the adaptive immune system and provides adaptive immunity to bacteria and archaea against foreign invaders such as plasmids and bacteriophages [68]. CRISPR and Cas proteins gather together to form a widespread system in bacteria and archaea, which interferes with foreign nucleic acids. The CRISPR–Cas system works in at least two stages: the adaptation stage, in which cells acquire new spacer sequences from exogenous DNA, and the interference phase, in which recently obtained spacers are used to target and cleave invasive nucleic acids. The CRISPR–Cas system participates in the continuous evolution of bacteriophages and bacteria by adding or deleting gaps in host cells and mutations or deletions in phage genomes [13,69].

### 2.4. Lack of Phage Pharmacokinetic Data

PT preparations are difficult to standardize. The definition of dosage remains unclear. In addition, the mode of administration and dosage of PT directly affects its effects, which results in difficulty in the clinical application of PT. Since bacteriophages are composed almost entirely of proteins and DNA or RNA, they can easily be degraded when they interact with human metabolism, such as in the stomach and liver, and when they are confronted by the animal immune system [70]. Related pharmacokinetic studies showed that a quarter of bacteriophage infusions lasted for 36 h after treatment, but their effective concentration was diluted by body fluids [71]. Oral administration was the most suitable mode for humans and animals. Furthermore, it was relatively easy and comfortable and induced low immunogenicity compared to other drug administration methods [72]. During oral administration, bacteriophage particles pass through the stomach, the intestine, and the intestinal mucosa before reaching the systemic circulation. Therefore, the gastrointestinal system is considered as the primary barrier in preventing tissue infiltration by bacteriophages [73]. In addition, the mammalian circulatory system effectively removes bacteriophages from the blood [74], which makes it difficult to maintain sufficient bacteriophage concentrations to destroy the target bacteria.

### 2.5. Interaction with the Body

Bacteriophages release bacterial toxins, such as endotoxins, when lysing bacteria, which worsens bacterial infections. In several cases, this even resulted in septic infections [75]. Related experiments showed that an oral phage cocktail in mice increased intestinal permeability and endotoxemia [76].

Foreign proteins carried by bacteriophages may induce immune reactions in humans or animals. This reaction is an exception, as bacteriophages have been shown to be safe; the reactions were allergic immune responses to phage virion-related proteins [77,78].

Numerous studies reported that PT was effective to treat a variety of diseases, but no data from double-blind randomized controlled clinical trials are available [79].

## 3. Solutions to the Limitations of PT

### 3.1. Solutions to Disadvantageous Bacteriophage Characteristics

The problem of a narrow host range can be solved in various ways such as by using phage mixtures [20], establishing a phage library [75], and performing extensive screenings [60]. A phage mixture is equivalent to a variety of drug therapies wherein different bacteriophages in a mixture can infect a variety of bacterial strains that may be present after a specific diagnosis [74]. McVay used a bacteriophage mixture to treat burned mice and were able to effectively reduce their mortality [80]. A phage library is a collection of isolated phages. These bacteriophages have certain characteristics and can be used as phage preparations or as unexpanded phage reserves to match recently isolated specific target bacteria [20]. The extensive screening of bacteriophages involves using a wide range of hosts to identify bacteriophages that employ common surface receptors to cleave a various pathogens, such as bacteriophages that target multiple isolates of two different pathogenic bacteria [81]. In addition to solving the problem of the host spectrum by taking advantage of a large number of bacteriophages, it can also help expand the host range of a single bacteriophage [74]. Expanding the host range of a single phage can be accomplished by using genetic engineering techniques to modify part of the phage responsible for host binding or by cloning a second alternative or additional version of these proteins involved in host binding into a single phage [82]. As a genetically modifiable biological nanoparticle, the T7 phage holds promise for biomedical imaging probes, therapeutics, drug and gene carriers, and detection tools [83]. In general, for the sake of efficacy, a phage mixture with more than one phage type to attack bacteria may be preferrable. On the other hand, for phage–host specificity, based on a direct matching with a specific target pathogen, the use of a phage library may be preferable [84].

One of the principles in preventing lysogenicity is to exclude temperate bacteriophages. PT must be achieved by lytic phages which must be highly purified to eliminate the immune effects of the infected lysogenic bacteria on similar bacteriophages.

Bacteriophages can encode enzymes that hydrolyze peptidoglycans, causing cell walls to degrade, thereby infecting host cells or releasing offspring viruses in host cells. The enzyme that causes peptidoglycan to dissolve from within is called endolysin. Through the application of endolysin in the treatment of bacterial diseases, the therapeutic effect is easy to evaluate compared with the direct use of bacteriophage therapy. This reduces the difficulty of quality evaluation. In recent years, studies developed the synthesis and transformation of lysin-coding genes into antimicrobial peptides [85], which greatly enhanced the antibacterial activity of the original bacteriophage.

### 3.2. Establishment of Relevant Policies and Standards

Meetings on the supervision and regulation of PT have been held several times in order to promote the development of PT [63,86]. Since the discovery of bacteriophages, bacteriophages have been widely used in Eastern Europe and the former Soviet Union; therefore, the use of therapeutic bacteriophages have been integrated into healthcare systems [87]. The open policy of PT facilitates its rapid development. Thus, relevant regulatory standards should be issued in time. In addition, bacteriophages are isolated and purified in different forms, but all methods involve similar steps, that is, environmental samples are collected and evaluated for the existence of bacteriophages. Standardized bacteriophage purification [88] has been repeatedly considered. Hence, a national standard scheme should be established for personalized PT. In addition, the standard operating procedure of the clinical application of PT should be clarified [89], including the recruitment of patients receiving PT, the establishment of phage libraries, the isolation and identification of pathogens, the screening of effective phages for pathogens, the preparation of phage formulations, management strategies, approaches to bacteriophage preparations, the monitoring of the efficacy of PT, and the detection of the emergence of phage-resistant strains.

### 3.3. Combined Dosage Regimens

In view of the emergence of anti-bacteriophage strains, bacteriophages can be used in combination with other antimicrobials, such as antibiotics. The combined use of bacteriophages with antibiotics is the best choice to address drug resistance and is also a step towards the transition from antibiotic therapies to PT, accelerating the development of the PT research industry. Numerous studies showed that a combination of bacteriophages and antibiotics was effective [90]. Related experiments showed that, compared with the separate use of linezolid, which is an effective inhibitor of protein synthesis, and a bacteriophage, the combination of the two minimized the adhesion of bacteria [91]. Bacteriophages and antibiotics have different bactericidal and bacteriostatic mechanisms. The synergistic effect produced by the combined use of bacteriophages and antibiotics not only helps restore the sensitivity of drug-resistant bacteria to antibiotics [92], but also reduces the probability of development of drug-resistant bacteria. The Chan test showed that phage selection restored the sensitivity of multidrug-resistant *Pseudomonas aeruginosa* to antibiotics [93]. Similarly, Oechslin et al. observed that the combination of a bacteriophage and ciprofloxacin showed a high degree of synergy in the treatment of experimental endocarditis in rats induced by *Pseudomonas aeruginosa* and effectively inhibited the emergence of anti-phage mutants [94]. Similarly, clinical cases reported that a wound healing preparation consisting of ciprofloxacin and bacteriophage polymers successfully treated patients infected with multidrug-resistant *Staphylococcus aureus* following radiation [26]. In addition, we know that biofilm pairs can improve the tolerance of bacteria to antibiotics, and several experiments showed that the combination of bacteriophages and antibiotics reduced the bacterial density in biofilms [95]. Certainly, several bacteriophages try to avoid bacterial resistance by expressing anti-CRISPR proteins that inhibit the resistance mechanism, overcoming the CRISPR–Cas immunity. To prevent the emergence of phage resistance, we should prioritize the use of phage mixtures [96] when using PT. Gayder found that competition and synergy between bacteriophages can be used to enhance the antibacterial efficacy of bacteriophage mixtures [97]. We should take advantage of the interactions between bacteriophages and bacteria to improve our battle against bacteria.

Antimicrobial peptides are innate immune components that exist in almost all organisms and have broad-spectrum antibacterial activity. Several studies showed that the combined use of bacteriolysin LysH5 and nisin against *Staphylococcus aureus* had a strong synergistic effect. Nisin enhanced lysin cleavage eight-fold [98,99]. The combination of drugs with different antibacterial mechanisms to deal with the problem of resistance to a single agent is a trend in the treatment of bacterial diseases.

### 3.4. Optimization of the Administration

The optimization of the drug delivery pathways needs to consider whether the bacteriophage can survive in the body and be transported to all parts of the body. If local infection is achieved through systemic circulation, the bacteriophage survives in a cycle long enough to reach the infected site. If the phage is provided in the intestine, it must survive until it enters the bloodstream. In view of the fact that many bacteriophages are sensitive to the low pH of the stomach, phages [100,101,102,103,104] are wrapped with a pH protectant. For instance, microencapsulation of phages in a natural biopolymer matrix [105] is used as a protective barrier against the gastric environment to reduce the inactivation of bacteriophages after ingestion; this ensures the efficacy of bacteriophages. Related experiments showed that the liposome-encapsulated phages can be effectively retained in the stomach and be protected until their release. When an encapsulated phage reaches the intestinal tract, adhesion to the intestinal wall temporarily protects them from bile salts and clearance through excretion [106]. In addition, it is also possible to increase the dose or titer of bacteriophages for a short time to prevent bacteriophage inactivation or loss due to antibodies before they reach the target bacteria [75,107].

### 3.5. Clinical Experience

The problem of endotoxemia caused by bacteriophage lytic bacteria occurs in specific circumstances involving life-threatening bacterial infections, rather than in general with antimicrobial therapy for bacterial infections. Therefore, as a special case, it can be expected that excessive bacterial lysis can be dangerous, and this situation must be prevented [75].

This allergic reaction is also rare because of the relative safety of bacteriophages. Thus, even if there is a reaction, it is mild when it occurs [77,78]. In the application of PT, it is necessary to observe the treatment objects in real time, record the dosage of phage and the symptoms due to adverse reactions, accumulate experience through a large number of clinical treatments, and thoroughly investigate the immune response bacteriophages may cause.

## 4. Summary

PT has some limitations, such as a narrow host range, lysogenicity, lack of relevant policies, lack of pharmacokinetic data, and so on, which have a certain impact on its clinical application. Table 2 lists and summarizes the limitations of PT, their clinical impact, and solutions. The limitations of PT mainly include three aspects. The first is the influence of the characteristics of the phage itself on the application of PT, the second is that there are no relevant laws, regulations, and standards for PT, and the last consists of the problems in clinical applications. It is essential for the further development of PT to review the relevant literature and formulate reasonable strategies to overcome these limitations.

## 5. PT Prospects

The goal of PT is to develop effective, rapid, and stable bacteriological drugs. The unique properties of bacteriophages make them highly competitive, and they are expected to be used in the treatment of infection of drug-resistant bacteria as a supplement to chemical antibiotics. After more than 100 years of boom and bust, PT has ushered in a new turning point in the context of drug resistance. A large number of experimental results show that phages are safe and effective. However, there are still some controversial issues to be resolved before phages can move to the clinical frontline, including ideal phage screening, effective dosage forms, and clinical practice. In relevant animal models, we need more ecophysiological data on the interaction between phages and bacteria in vivo in order to select suitable phages for clinical application. An ideal bacteriophage for therapeutic use should have strong cleavage ability, good environmental adaptability and stability, no endotoxin gene in the genome, a relatively wide cleavage range, be easy to isolate and purify, and no negative effects on the human immune system. In this era of rapid emergence and spread of drug-resistant bacteria, it may be necessary to further understand phage biology and establish scientifically effective pharmaceutical standards for phages, so that they can have a second chance and receive the same attention as antibiotics. It is believed that with the increase in phage research and the progress in clinical trials, these limitations can be resolved.

## Figures and Tables

**Figure 1 molecules-27-01857-f001:**
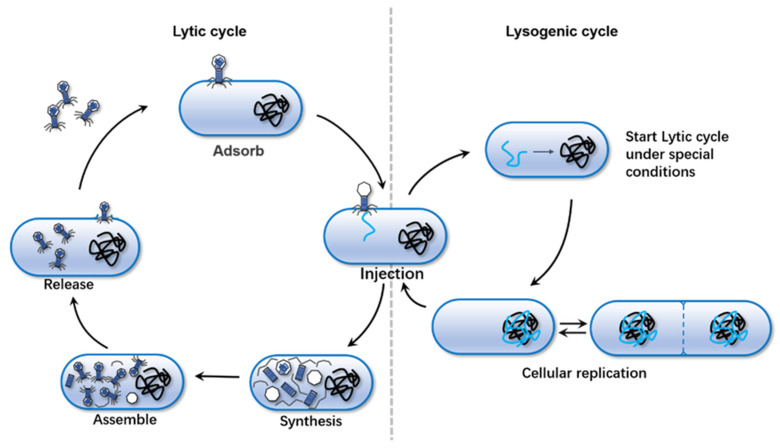
Phage lysis cycle and lysogenic cycle.

**Table 1 molecules-27-01857-t001:** Application field of phage therapy and corresponding host bacteria.

Application Field		Phage Treatment	Host Bacteria	Treatment Effect	Reference(s)
Human		Oral phage mixture (including 676/F, A3/R and A5/80)	Resistant *Staphylococcus aureus*	Successfully decolonized drug-resistant *S. aureus*	[25]
		The film covers the wound surface and contains PhagoBioDerm, a new type of slow-release biopolymer impregnated with lysophage, antibiotics, and analgesics		Negative *S. aureus* test, wound healing	[26]
		A filter paper disc soaked in a purified phage suspension covers the infected area	*Pseudomonas aeruginosa*	Three days after applying the phage, *P. aeruginosa* was not isolated from the swab	[27]
Animal	Cattle	Injection of phage Ø26, Ø29, Ø21, Ø27, Ø6, Ø44, Ø16, Ø39, Ø55, and Ø51	Shigatoxigenic *Escherichia coli*	Improved calf diarrhea, lowered rectal temperature, and increased calf weight	[28]
	Poultry	Cloaca drops or oral bacteriophages CB4Ø and WT45Ø	*Salmonella* Enteritidis	Significantly reduced *Salmonella* Enteritidis in cecal tonsils	[29]
		Aerosol spray or drinking water to administer phage BPs * (mixture of phage BP1, BP2 and BP3)		Reduced *Salmonella* infection incidence and number in the intestine	[30]
		Oral or spray administration of BPs and Broilact (a commercial probiotic product)		Significantly reduced *Salmonella* infections in cecal samples	[31]
		Oral bacteriophages CNPSA1, CNPSA3 and CNPSA4		[32]
		Oral phage ØCJ107		Significantly reduced colonization and horizontal spread of *Salmonella*	[33]
		Oral bacteriophages S2a, S9 and S11 and Protexin (a probiotic product)		Significantly reduced the number of *Salmonella typhimurium* in the liver, spleen, ileum, and cecum of chicks	[34]
		Spray and intramuscular injection of bacteriophage SPR02 and DAF6		Significantly reduced mortality	[35]
		Bacteriophage SPR02 and DAF6 injection		Significantly reduced mortality rate and incidence and severity of air sacculitis injury	[36]
		Bacteriophage SPR02 airbag inoculation		Significantly reduced mortality	[37]
		Mixture of phages phi F78E, phiF258E, and phi F61E		Significantly reduced morbidity and mortality	[38]
		Aerosol spray of bacteriophage SPRO2 and DAF6		Significantly reduced weight loss and mortality	[39]
	Pig	Mixture of phages F3, F4, F5, F6, F7, and F8	*Salmonella* Typhimurium	Significantly reduced *Salmonella typhimurium* colonization	[40]
		Oral two-strain phage mixture		Reduced intestinal colonization of *Salmonella typhimurium*	[41]
		14 kinds of phage mixture (PEW 1–14) gavage and oral administration, microencapsulated		Reduced *Salmonella* colonization	[42]
		Oral microcapsules composed of 14 kinds of phage mixtures (PEW 1–14) and bacteriophage Felix O1		Reduced colonization of *Salmonella typhimurium* in the ileum, cecum, and tonsils of pigs	[43]
		Oral phage CJ12	*E. coli*	Decreased diarrhea rate and significantly reduced *E. coli* abundance in feces	[44]
		Oral mixture of several of the 7 phages (GJ1–GJ7)		Prevention or treatment of diarrhea, significantly reduced damage by diarrhea	[45]
	Mouse	Intraperitoneal injection of bacteriophages ENB6 and C33	*Enterococcus faecalis*	Significantly reduced mortality	[46]
		Intraperitoneal injection of phage ØEF24C		Significantly reduced mortality	[47]
		Intraperitoneal injection of a single dose of phage CSV-31		Significantly reduced mortality	[48]
		Intraperitoneal injection of phage ØA392		Significantly increased survival rate	[49]
		Intraperitoneal injection of bacteriophage SS	*Klebsiella pneumoniae*	Significantly reduced *K. pneumoniae* bacteria in the lung tissue	[50]
		Intraperitoneal injection of phage ØNK5		Significantly inhibited liver damage and death caused by *K. pneumoniae*	[51]
Plant	Citrus fruit trees	Spray phages CP2, ØXac2005-1, ccØ7, ccØ13, ØXacm2004-4, ØXacm2004-16, ØX44, ØXaacAl	*Xanthomonas axonopodis*	Significantly reduced the severity of citrus canker and citrus bacterial spot	[52]
	Potato	Phage φMA1, φMA1A, φMA2, φMA5, φMA6, and φMA7	*Pectobacterium carotovorum, P. atrosepticum*,	Significantly reduced rate and area of soft rot	[53]
Food	Beef/vegetables and ground beef	Use of a dropper to administer a mixture of the phages e11/2, pp01, and e4/1c, dropwise	*E. coli*	Eliminated or significantly reduced *E. coli* abundance	[54]
		Coated phage ECP-100 (a mixture of ECML-4, ECML-117, and ECML-134)		Significantly reduced the number of *E. coli* on the surface of vegetables and ground beef	[55]
		Dip into the washing solution made of bacteriophages C14, V9, L1, and LL15		Significantly reduced *E. coli* abundance on vegetables	[56]
	Chicken skin/fresh cut fruit/sausage	Spray phage type 4 strains P125589, P22, and 29C	*Salmonella*	Significantly reduced the number of *Salmonella* on the surface of chicken skin	[57]
		Drops of bacteriophage Felix O1		Significantly suppressed the number of *Salmonella* in sausages	[58]

* BP: Bacteriophage.

**Table 2 molecules-27-01857-t002:** Limitations of PT, their impact on clinical applications, and solutions.

Types of Defects in Bacteriophage Therapy	Limitations of Bacteriophage Therapy	Implications for Clinical Applications	Solution
Disadvantages of bacteriophages	Phage specificity	Unable to treat mixed bacterial infections	Phage Mix
	Lysogenic phage	Cannot lyse bacteriophages and may transmit toxin genes and drug resistance genes to bacteria	Strict use of lytic phage
	The composition is complex, and the quality and efficacy test and evaluation are difficult	Unable to assess its quality and efficacy	Determine the dosage form and concentration of phage preparations, compare similar drugs, and select reasonable evaluation methods
	Bacteria resistance to bacteriophages	Lead to ineffective treatment	Combined dosage, regimens with antibiotics or probiotics, phage mixtures
Policies, regulations and standards	Lack of regulations and policies	Lack of regulatory supervision, easy to abuse	Formulate regulations and improve policies
	Lack of separation and purification standards	The isolated phage is not standardized enough and not pure enough	Establish complete separation and purification standards
Clinical application	Determination of administration method and dosage form	Different dosage forms affect the efficacy	Explore the advantages and disadvantages of different administration methods and dosage forms
	Lack of pharmacokinetic data	Unable to determine the half-life and action time of the phage in the body	Statistic data on pharmacokinetics of different formulations of phage and use of phage protectors
	Endotoxin release	May cause endotoxemia	Establish a treatment plan for the foreseeable release of endotoxin
	Phage protein immune response	May cause immune stress	Record possible immune response through a large number of clinical trials
	Lack of data from double-blind randomized controlled clinical trials	Unsure of its efficacy	Double-blind randomized controlled trials of phage therapeutics to evaluate their therapeutic effects

## Data Availability

Data are contained within the article.

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
