# Peer review of "Limitations of Phage Therapy and Corresponding Optimization Strategies: A Review"

_molecules, 2022, doi:10.3390/molecules27061857_

Round 1

Reviewer 1 Report

The review paper Limitations of phage therapy and corresponding strategies fits well with the aims of the journal.

It recalls the basic biology of phage replication and the interest of the main developments of using phages in therapeutic approaches. By providing an overview of the variety of applications ranging from clinical work in diseases caused by bacteria in humans, in animals but also in agriculture/plants, the paper covers the main attempts of such developments of phage therapies. This provides indirectly the information of the spectrum of successful applications of phage therapy during the past decades/ century.

Comments:

In the 2nd main paragraph, the authors point well the limitations and difficulties of the developed approaches in phage therapy.

It could be recalled that many of these therapeutic approaches were mainly developed in experimental (also clinical) pioneer work. The considerations are mainly based on the biology of the phages. They reveal the limitations considering both the biological circumstances / interaction of the phage and host bacteria and the pharmacological interaction with the treated organism (mainly human), and in consequence the missing standardized therapeutic evaluation and the missing regulatory framework in such applications.

The 3rd paragraph reviews constructive approaches to face the spectrum of the observed limitations by providing potential solutions. In that sense the authors recall and provide potential solutions to handle disadvantageous bacteriophage characteristics, to propose bio-pharmacology considerations, and the necessity of an administrative framework for the rapid development of efficient bacteriophage based anti-bacteriological drugs in perspective of clinical application.

Title of 4th paragraph could be renamed : Impact of limitations of PT on clinical applications (instead of Summary)

Overall:

The paper is well written and easy to follow. The provided comments are very informative, especially for the readers that are not familiar with the therapeutic potential and implicit considerations of phage biology.

Minor: Table 1 needs reorganization by positioning the headers differently.

Author Response

Thank you for your comments. We have revised the article accordingly.

Minor: Table 1 needs reorganization by positioning the headers differently.

Reply:I have made adjustments to the table positioning in the text.

Reviewer 2 Report

The title is incomplete and should explicitly specify what strategies are being referred to.

The introduction background refers more to antibiotic resistance than phage therapy and should be modified to highlight the knowledge gap from literature. The review article presents some of the limitations experienced with using phage therapy as a treatment option for bacterial resistance. The limitations and potential solutions are discussed broadly and span randomly over a number of years. This could be refined/ described in sub-categories according to specific timeframe. The information presented in the tables can be further elaborated on.

The limitations should be described for the past five years and the solutions recommended are potential/ temporary since most of the information presented in the review is basic with literature cited up until 2021. There is no updated information presented from 2022 which is available online.

This review article should follow a scoping review format.

Extensive editing of the English language and style is required.

Author Response

Thank you for your comments. We have revised the article accordingly.

1.The title is incomplete and should explicitly specify what strategies are being referred to.

Reply:

Thank you for your valuable comments. Regarding the "strategies" mentioned in the title, the text includes "1. Solutions to disadvantageous bacteriophage characteristics, 2.Establishment of relevant policies and standards, 3.Combined dosing regimen, 4.Optimization of administration, 5.Clinical experience", I don't think a specific solution can be added to the title. I will change the title to "Limitations of phage therapy and corresponding optimization strategies" after listening to your opinion.

2.The introduction background refers more to antibiotic resistance than phage therapy and should be modified to highlight the knowledge gap from literature. The review article presents some of the limitations experienced with using phage therapy as a treatment option for bacterial resistance. The limitations and potential solutions are discussed broadly and span randomly over a number of years. This could be refined/ described in sub-categories according to specific timeframe. The information presented in the tables can be further elaborated on.

Reply:

Thank you for your valuable comments, the background of this article includes 1.1 Background, 1.2 Bacteriophage, 1.3 Phage therapy (PT), 1.4 PT application, I have done a specific description of phage therapy in 1.3, I think it is my subtitle question right You are misleading, and at your suggestion I have changed the title of 1.1 to "1.1 Status of bacterial resistance".

I have detailed the tables section of the article.

3.The limitations should be described for the past five years and the solutions recommended are potential/ temporary since most of the information presented in the review is basic with literature cited up until 2021. There is no updated information presented from 2022 which is available online.

Reply:

Thank you for your valuable comments. Regarding your comments, I have added relevant literature in 2022 to the text.

4.This review article should follow a scoping review format.

Reply:

Thank you for your valuable comments, some changes have been made in the text.

5.Extensive editing of the English language and style is required.

Reply:

Thanks for your valuable comments, the sentences and errors have been corrected in the text.

Round 2

Reviewer 2 Report

The authors have responded to my previous comments, and have included the required information.

This manuscript is a resubmission of an earlier submission. The following is a list of the peer review reports and author responses from that submission.